# Language-Independent Fake News Detection: English, Portuguese, and Spanish Mutual Features

**Hugo Queiroz Abonizio** [1,†] , **Janaina Ignacio de Morais** [1,†] , **Gabriel Marques Tavares** [2,†]
**and Sylvio Barbon Junior** [1,*,†]

1   State University of Londrina (UEL), 86057-970 Londrina, Brazil; hugo.abonizio@uel.br (H.Q.A.);
    janainam@uel.br (J.I.d.M.)
2   Università degli Studi di Milano (UNIMI), 20122 Milan, Italy; gabriel.tavares@unimi.it
*   Correspondence: barbon@uel.br
†   These authors contributed equally to this work.

**Abstract:** Online Social Media (OSM) have been substantially transforming the process of spreading news, improving its speed, and reducing barriers toward reaching out to a broad audience. However, OSM are very limited in providing mechanisms to check the credibility of news propagated through their structure. The majority of studies on automatic fake news detection are restricted to English documents, with few works evaluating other languages, and none comparing language-independent characteristics. Moreover, the spreading of deceptive news tends to be a worldwide problem; therefore, this work evaluates textual features that are not tied to a specific language when describing textual data for detecting news. Corpora of news written in American English, Brazilian Portuguese, and Spanish were explored to study complexity, stylometric, and psychological text features. The extracted features support the detection of fake, legitimate, and satirical news. We compared four machine learning algorithms (k-Nearest Neighbors (*k*-NN), Support Vector Machine (SVM), Random Forest (RF), and Extreme Gradient Boosting (XGB)) to induce the detection model. Results show our proposed language-independent features are successful in describing fake, satirical, and legitimate news across three different languages, with an average detection accuracy of 85.3% with RF.

**Keywords:** fake news; text classification; multi-language; stylometry; machine learning

## 1. Introduction

The way to deliver and consume information has changed significantly today. Internet access has become more democratic and fast, paving the way to spreading the news around the world in seconds. In 2017, the Brazilian Institute of Geography and Statistics (IBGE) published a survey (https://biblioteca.ibge.gov.br/visualizacao/livros/liv101631_informativo.pdf) regarding the use of the Internet by Brazilians, of which 95.5% access the global network to send and receive messages through Online Social Media (OSM). Furthermore, a survey (https://www.statista.com/chart/15355/social-media-users) by Statista indicates that, until 2021, more than one-third of the globe will be connected via OSM, showing how access to information will become available and affordable in the future.

In recent years, OSM grew into one of the most popular communication technologies for various types of personal relationships [1,2]. Most people expose their opinions, talk to loved ones, and share professional information and news about the world [1,3]. Thus, it is common to quickly find accurate opinions on the same subject, which enables an increase of critical and abstract thinking on current issues.

Traditionally, news was disseminated by influential newspapers and television media only. Today, however, news articles can be written and spread by anyone with access to the Internet. Moreover, OSM promote channels that do not have editorial norms nor content review, which can be a problem since some people might suffer from a lack of critical analysis of news [4].

Shao et al. [5] highlighted as a natural human behavior the absence of concern in verifying the credibility of most of the information given in blogs and virtual encyclopedias, and the natural tendency to believe in the content shared. For example, the 2016 American elections were pervaded by the massive dissemination of information through OSM, which biased the voting decisiin Reference [6]. However, not all information was confirmed and verified, which brings us to one of the most severe current issues, the fake news.

Fake news is similar in appearance to legitimate news [7] but refers to news articles created to deceive the reader, either for the author benefit or that of a third party, generally involving monetary gain Reference [6]. Shu et al. [8] define the fake news detection problem as a function $F(a)$, where $F$ is a prediction function and $a$ is a news article. Given a news article $a$, which is described by the *Publish* and *Content* set of attributes, $F$ predicts if $a$ is a fake news piece or not. Although this approach defines fake news detection as a binary classification, other works have addressed several types of fake news [9–11].

Rubin et al. [9] describe three types of fake news: *serious fabrications*, which are deliberate fraudulent reports, *large-scale hoaxes*, which is another type of falsification that may be mistakenly validated by traditional media, and *humorous fakes*, where readers are aware of the humorous intent. Salas-Zárate et al. [11] explore the dichotomy between satirical and non-satirical news, while Rubin et al. [10] develops a link between deception detection and computational satire, irony, and humor detection.

Currently, several websites, like Sensacionalista (https://www.sensacionalista.com.br/), Actualidad Panamericana (https://actualidadpanamericana.com/), and The Daily Discord (http://www.dailydiscord.com/), use humor to create satirical news related to some subject in an exaggerated way, making clear to the reader that this information is not legitimate [11]. However, satirical news can be shared on social networks or suspicious sites without its original context, hence creating the possibility of deceiving the most distracted readers [10]. This happens because the satire uses a format very similar to traditional journalism, which, leveraged by an out-of-context sharing, can be confused as a real story [11]. Moreover, there are several specific strands of various languages that compromise automatic detection, for example, misleading news containing partial truthful informatiin Reference [6]. Therefore, it is not an easy task to identify whether a news article is fake, satirical, or legitimate [11,12].

Approaches on fake news detection can be split into two categories [8]: social context-based and content-based. The social context-based approaches usually analyze the propagation patterns and the diffusion on social networks to identify deceptive content. The content-based approach can be further divided into two types: knowledge-based, which uses knowledge databases to verify the information; and style-based, which extracts writing style and linguistic features to detect deception. The knowledge-based approaches often use public structured knowledge linked data, as described in Conroy et al. [13].

Initially, social context-based solutions seem the most adherent to address the problem of fake news spreading in a language-independent way. However, the requirements of meta-information about the structure, path, and distribution pose several disadvantages to these solutions. Indeed, the complexity demanded by this category to support suitable results paves the way to content-based solutions.

In recent years, several content-based solutions have been proposed, mainly based on Natural Language Processing (NLP) and Text Processing (Text Mining). From detecting fake news through a pure NLP [14] classifier, to even distinguishing satire from fake news using social networks as seen in

Reference [10]. However, one important challenge of fake or satirical news detection is related to the limitation of language-independent written features [15].

Most content-based works, mainly from linguistic approaches, are based on specific languages, such as, English [16], Spanish [11], and Chinese [17]. For instance, Pilar et al. [11] analyzed satirical news from the Twitter (https://twitter.com/) social network coming from two Spanish-speaking countries: Mexico and Spain. Their goal was to find differences and linguistic similarities of news that indicate sarcasm.

Meanwhile, in other NLP tasks, such as Sentiment Analysis, there is an effort to address the challenge of a language-independent approach, as shown in Kincl et al. [18]. Thus, to the best of our knowledge, there are no fake news detection techniques, nor satirical ones, that handle more than one language, showcasing the lack of a general approach.

Therefore, the main goal of this work is to propose and compare language-independent features to detect news considering three classes: fake, satirical, and legitimate. For that, we built a pipeline using content-based premises to retrieve news style by extracting *Complexity*, *Stylometric*, and *Psychological* features in Brazilian Portuguese, American English, and Spanish textual data.

The remainder of this paper is organized as follows. In Section 2, we present a review of related work, comparing their methods with the methodology being proposed in this paper. The methodology of experiments is described in Section 3, detailing the source of the dataset, the extracted features, and the Machine Learning (ML) algorithms employed. The results of the experiments are discussed in Section 4, detailing the behavior of the most important features and the shared characteristics between datasets. Finally, we conclude the paper in Section 5.

## 2. Definitions and Related Work

Recently, there has been an increasing amount of literature on fake news detection tasks. This paper uses three categories for news articles: fake, legitimate, and satirical. The formal definitions used in this work are extracted from Shu et al. [8]:

**Definition 1** (Fake News). *Let a news article A, described by a tuple of two features: authenticity and intent. A is considered fake if the authenticity is verifiable false, and its intent is to mislead the reader, e.g., its content is deceptive.*

**Definition 2** (Legitimate News). *Let a news article A, described by a tuple of two features: authenticity and intent. A is considered legitimate if the authenticity is verifiable true, and its intent is to convey authentic information to the reader, e.g., its content is reliable.*

**Definition 3** (Satirical News). *Let a news article A, described by a tuple of two features: authenticity and intent. A is considered satirical if the authenticity is verifiable false, and its intent is entertainment-oriented and reveals its deceptiveness to the consumers.*

Several datasets have been proposed to serve as benchmarks. They vary in number of samples, availability of content, and source languages, but there is still no consensus used by most papers of the field. The *LIAR* dataset, proposed by Wang [19], contains more than 12,000 human-labeled short statements with fine-grained gradations of truthfulness. However, this dataset composed by short statements, making it difficult to use style-based approaches to identify deception as this type of approach requires more content information. Thus, knowledge-based solutions are more fit in this case. Hanselowski et al. [20] and The fake news Challenge [21] published datasets with a similar proposal: given a claim, the system predicts if other statements are mainly agreeing or disagreeing with it. Though such datasets have a considerable amount of samples, the task is beyond the scope of the present study. To overcome this, we focused on datasets with complete news to classify documents into one of the three categories being assessed in this work, selecting equivalent corpora from different languages.

Horne and Adali [22] assembled collected news written in English from different sources and made the corpus publicly available. The authors use a classifier based on writing style features to identify the document's class. The set of features takes into account writing characteristics, such as the frequency of grammatical classes and readability measures. However, many features are language-dependent, which limits the method's applicability. We use their proposed corpus in this work, but, due to the low number of instances, it was complemented. The process of samples complementing is described in Section 3.2.

Zhou et al. [23] explored possible patterns in fake news and its potential relationship to deception and clickbait. For this, the authors proposed a model focused on the theory of detecting false news, where a news story is investigated at the lexical, syntactic, and semantic levels, which differs from approaches that focus on news content. The authors found that the proposed method can overcome the state-of-the-art, in addition to allowing early detection of false news, even when there is some limitation on the content. However, the results obtained did not consider possible divergences in linguistic structures and did not address any language other than English.

Monteiro et al. [24] and Posadas-Durán et al. [25] developed and made publicly available, respectively, the corpus Fake.Br, written in Portuguese, and FakeNewsCorpusSpanish, written in Spanish. Both works proposed text classifiers that use textual features. However, some features used on those works, such as Bag-of-Words, are language-dependent. Another difference between those works and ours is that the former treats the problem as a binary classification (legitimate and fake), while we treat as a multi-class problem (legitimate, fake, and satirical), understanding that satirical news is a separate type of document.

Morais et al. [26] proposed a Decision Support System (DSS) based on a multi-label text classification pipeline for news into two conceptual classes: objective/satirical and legitimate/fake. For this, the authors used a Portuguese dataset collected from Brazilian sites and considered the stylistic features from the news in DSS. As a result, news can be categorized as objective and legitimate, satirical and fake, or any other combination of those two classes at the same time. Nonetheless, the work is limited to a Portuguese dataset with features proposed and evaluated only on the Portuguese language structure. Different from Reference [26], in this work, we considered different languages and evaluated the extracted features in a language-independent setup.

Krishnan and Chen [27] and Sousa et al. [28] focused on identifying fake news spread directly from social networks, specifically Twitter in those cases. The usage of Deep Learning (DL) methods is also present on the field, where Rashkin et al. [16] applied a Long Short Term Memory (LSTM) model to obtain gradual authenticity score of the news. However, DL methods usually depend on large amounts of data to extract patterns. Therefore, traditional ML methods fit better on real scenarios, where data availability is limited.

Gruppi et al. [29] collected datasets in two languages (Portuguese and English) and analyzed the similarities in stylometry in both languages, based on the search for universal characteristics that are independent of culture, or specific attributes to each language. The authors found that the attributes of unreliable articles follow a similar pattern in both languages, suggesting the existence of stylistic characteristics when separating reliable and unreliable articles in both languages. However, the results were restricted to only two languages, not enough to conclude if stylometric patterns are observed in multiple languages. In addition, the authors only inspect the reliability of the articles, without necessarily verifying the veracity of the information or analyzing other characteristics, such as satire. Guibon et al. [30] addressed a dataset composed of both French and English to distinguish fake, trusted and satire contents, but, different from our proposal, their data representation remain language-dependent, since it relies on specific languages pretraining methods like word vectors or term frequencies.

Therefore, given the lack of study on fake news detection on languages other than English, a comparison between different languages is one of the main contributions of this work. Moreover, previous studies have evaluated only fake and legitimate news, while we leveraged a more broad

scenario by considering fake, legitimate, and satirical news. Instead of adhering to a corpus collected on a single language, three corpora of distinct languages (English, Portuguese, and Spanish) were used to conduct the experiments. The models were induced in each corpus, evaluated, and the most important features were compared by analyzing the characteristics shared among languages. The features used in this paper are carefully proposed and chosen to be language-independent, in order to test the same set of features on different corpora idioms, increasing the applicability of our method.

## 3. Material and Methods

This section presents the proposed (i) text processing pipeline, (ii) language-independent features, (iii) news collection process, and (iv) classification algorithms. For this study, we created a dataset composed by news documents written in American English, Brazilian Portuguese, and Spanish to evaluate the problem of fake news detection grounded on language-independent features. The features were extracted from three corpora and had their importance evaluated in the detection task. Moreover, the designed features model high-level structural text characteristics, rather than the specific words contained in the text.

### 3.1. Text Processing Pipeline

Figure 1 shows an overview of the pipeline composed of three steps: Preprocessing, Feature Extraction, and Classification. The pipeline covers the handling of raw news until the detection outcome. Considering a news article has different sources, the proposed features were carefully created to avoid capturing metadata characteristics, e.g., the amount of spacing, web template, and specific publisher style. It is important to highlight the feature vector is composed by our language-independent proposed features together with traditional descriptors from fake news detection literature.

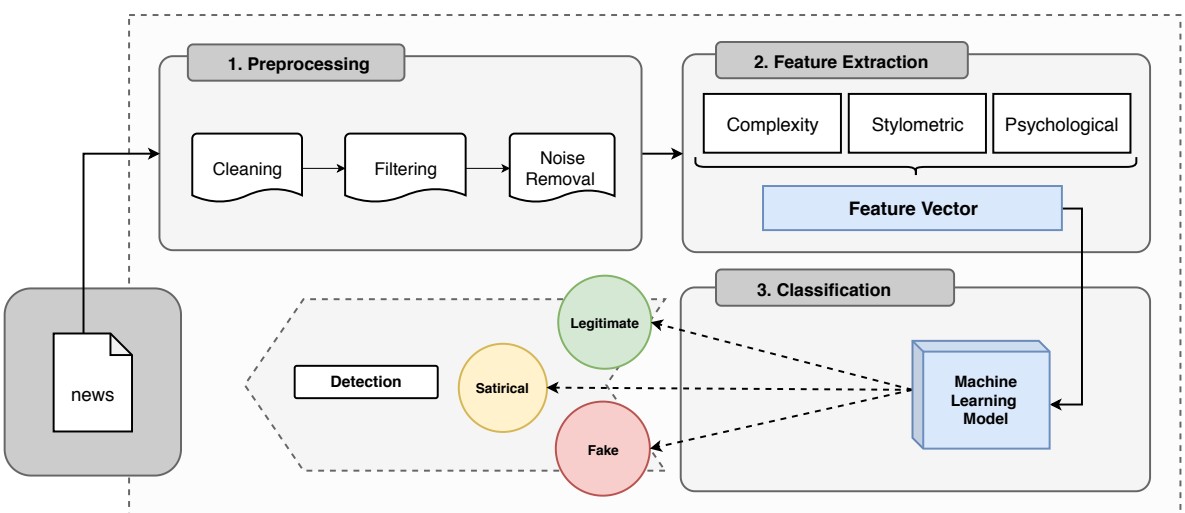

**Figure 1.** Overview of pipeline to detect news articles among fake, legitimate, and satirical using language-independent features

In the first step, Preprocessing (1), a set of procedures are performed to eliminate unwanted characteristics left by the data acquisition phase. The Cleaning phase converts all samples into UTF-8 encoding format and removes non-textual characters, such as *emojis* and special characters. This manipulation clears website metadata and leaves only news-related textual data. Then, in the Filtering phase, small texts are filtered out from samples to avoid news that are too short. Such process creates a homogeneity in text lengths, listed as a requirement for fake news detectiin Reference [9]. Lastly, the Noise Removal phase removes extra whitespaces, as well as normalizes quotes characters and other collecting and processing related noise that are not related to the news content.

After preprocessing the textual data, the news articles have their features computed in the Feature Extraction (2) step. The proposed features are language-independent, i.e., they do not consider specific terms from a language. Instead, the features focus on capturing high-level structures. This way, the same set of features can be used in multi-language domains. To achieve this goal, we extracted features from three categories, based on Horne and Adali [22] categorization: Complexity, Stylistic, and Psychological.

Complexity-based features aim at capturing the overall intricacy of the news, both in the sentence and word level. For that, we used metrics, such as average word size, words count per sentence, and Type-Token Ratio. Stylistic features use NLP techniques to extract grammatical information from each document, understanding its syntax and text style. Hence, part of speech (POS) tagger is used to track different word style frequencies. Psychological features are mostly related to cognitive processes. For that, we evaluated sentiment polarity [31], which measures the negativity or positivity of a text. Table 1 lists all features used in this paper. Some features have been proposed by other works, but they were never evaluated in a multi-language setup. Thus, we decided to include them in our feature vector to investigate their applicability in scenarios with more than one language.

**Table 1.** List of extracted features. There are three types of features: Complexity, Stylometric, and Psychological. POS = part of speech; OOV = Out-Of-Vocabulary, ADJ = Adjectives, ADP = Adposition, ADV = Adverbs, DET = Determiner, NOUN = Noun, PRON = Pronoun, PROPN = Personal Pronouns, PUNCT = Punctuation, SYM = Synonyms and VERB = Verb.

| No | Type | Name | Description | Reference |
|----|------|------|-------------|-----------|
| 1 | *Complexity* | words_per_sents | Average words per sentence | [22,32] |
| 2 | *Complexity* | avg_word_size | Average word size | [32,33] |
| 3 | *Complexity* | sentences | Count of sentences | [34] |
| 4 | *Complexity* | ttr | Type-Token Ratio (lexical diversity) | [32,35] |
| 5 | *Stylometric* | pos_diversity_ratio | POS-tag diversity | Proposed |
| 6 | *Stylometric* | entities_ratio | Ratio of Named Entities to text size | Proposed |
| 7 | *Stylometric* | upper_case | Uppercase letters | [36] |
| 8 | *Stylometric* | oov_ratio | OOV words frequency | Proposed |
| 9 | *Stylometric* | quotes_count | Quotation marks count | [22] |
| 10 | *Stylometric* | quotes_ratio | Ratio of quotation marks to text size | Proposed |
| 11 | *Stylometric* | ratio_ADJ | ADJ tag frequency | [22] |
| 12 | *Stylometric* | ratio_ADP | ADP tag frequency | [22,32] |
| 13 | *Stylometric* | ratio_ADV | ADV tag frequency | [22] |
| 14 | *Stylometric* | ratio_DET | DET tag frequency | [22] |
| 15 | *Stylometric* | ratio_NOUN | NOUN tag frequency | [22,32] |
| 16 | *Stylometric* | ratio_PRON | PRON tag frequency | [22,32] |
| 17 | *Stylometric* | ratio_PROPN | PROPN tag frequency | [22] |
| 18 | *Stylometric* | ratio_PUNCT | PUNCT tag frequency | [22] |
| 19 | *Stylometric* | ratio_SYM | SYM tag frequency | [22] |
| 20 | *Stylometric* | ratio_VERB | VERB tag frequency | [22,32] |
| 21 | *Psychological* | polarity | Sentiment polarity | [22,36] |

For Complexity features, documents were split into sentences and tokens through a tokenization process. Thus, we can extract the average words per sentence, average word size and the total number of sentences. Complexity metrics are inspired by readability indexes, such as Simple Measure of Gobbledygook (SMOG) Grade [37] and Automated Readability Index [38], which use word and sentence level measures to quantify a reading difficulty score. Type-Token Ratio (TTR) is also extracted by counting the number of unique words divided by the total number of words, measuring the vocabulary variation of the document. Those textual statistics are intended to help the characterization of the complexity of differences between news classes. Word and sentence level features are explored in the literature, as seen in Reference [8,22,24,32,33].

The Stylometric features take into account more advanced NLP techniques to extract grammatical and semantic characteristics from the text. The Polyglot package was used as a Named Entity Recognition (NER) [39] to detect named entities, counting recognized entities and their ratio to total text size. A similar approach was proposed by Rubin et al. [10], but the authors used as a feature to detect absurdity on news.

For POS-tagging, the spaCy package was used to label each token. The package has pretrained models for a variety of languages, making it possible to extend this work to other languages. The tags were used to extract a ratio of specific tags (e.g., VERB) to the total amount of tokens in the document. From that, we can assess the POS-tag diversity, i.e., the ratio of different tags present on text to total text size. Along with POS-tag features, the number of quotation marks and uppercase letters are computed, as proposed in Reference [22]. Moreover, we proposed the usage quotation marks frequency as a feature.

Furthermore, we proposed the usage of an Out-Of-Vocabulary (OOV) feature, which uses a dictionary of words for a given language and counts the total words that are not found in this set and their frequencies on the text. The hypothesis behind this feature is to capture neologisms, slangs or other kinds of unusual words. To compute this feature, only the words tagged as adjectives, adverbs, verbs, and nouns were considered. Some features were ignored due to its low variability on at least one of the datasets (ratios of AUX, CONJ, SCONJ, CCONJ, CONJ, PART, and INTJ tags to text size). We disregard those which standard deviation was close to zero as non-informative features.

The sentiment polarity was extracted using Polyglot [40], a Python package which implements multi-language sentiment lexicon, as described by Chen and Steven [41]. The package implements sentiment polarity extraction for several languages, including the ones evaluated in this study, which makes it possible to extend the application to other languages.

Therefore, the proposed language-independent features were carefully crafted to represent textual characteristics from several perspectives. Such a plural feature vector, as described in Section 3.3, supports the induction of ML models, which then perform the Classification step (3). The outcome of the pipeline is the decision result, which assigns one of the three classes, legitimate, satirical or fake, to a news article.

### 3.2. News Datasets

The experiments conducted in this study include documents originally written in American English (EN), Brazilian Portuguese (PT), and Spanish (ES). For that, we created a corpus composed of news articles proposed in other works and complemented them when necessary. Namely, we used: the corpus of Horne and Adali [22] for EN news; the Fake.Br corpus [24] for PT news; and the FakeNewsCorpusSpanish corpus [25] for ES news. Note that Reference [24,25] used only fake and legitimate classes, without considering the satirical class.

As the original corpora had an uneven distribution of classes (fake, satirical, and legitimate), we augmented them in an effort to standardize the news collection used in our experiments. To achieve this goal, we increased the number of samples by adding documents extracted from the Fake News Corpus (FNC) FNC is an open source dataset composed of millions of news articles collected on OpenSources ), a list of credible and non-credible online sources. The list uses several tags to label documents. We considered documents tagged as *Fake News*, *Satire*, and *Credible* categories, which correspond to fake, satirical, and legitimate classes, respectively. Moreover, FNC contain news articles written in both ES and EN. To detect the language employed in the document, we used the Python package *whatthelang* [42].

Since the FakeNewsCorpusSpanish lacks satirical news, we complemented this collection with FNC samples. For that, we selected ES samples with the *Satire* label from FNC. Regarding the EN collection, the corpus from Reference [22] consists of only 326 documents. To complement it, FNC documents of *Fake News*, *Satire*, and *Credible* were added to the EN corpus. The FNC does not include news written in PT. Thus, to overcome this gap, following the collecting method in Reference [26],

we extracted news from two widely known Brazilian satirical portals: Sensacionalista and Diário Pernambucano. Therefore, the corpora used our work is backed both by literature and open source community datasets.

Lastly, due to classes imbalance, sampling was made in order to keep the same number of documents for each class in order to avoid the class imbalance problem [43,44]. The experimentation was carried out using 9930 news articles split into three datasets grouped by language. As shown on Table 2, the EN corpus contains 2043 samples of each class, for a total of 6129 documents, while the PT corpus contains 846 samples of each class, totaling 2538 documents, and ES contains 421 samples for each class and 1263 in total. The EN corpus has 4,432,906 tokens, of which 94,496 are unique (2.1%), and an average of 723 tokens per document. While the PT corpus contains 1,246,924 tokens, of which 58,129 are unique (4.6%), with an average of 491 tokens per document. Lastly, the ES corpus contains a total of 459,406 tokens with 40,891 of them being unique (8.9%), with an average of 364 tokens per document. Table 3 shows examples for each class on each language.

**Table 2.** Corpora information summarising characteristics for English, Portuguese, and Spanish datasets. FNC = Fake News Corpus.

| Corpus | Samples | Samples per Class | Tokens | Unique Tokens | References |
|---|---|---|---|---|---|
| **English (EN)** | 6129 | 2043 | 4,432,906 | 94,496 | [22], FNC |
| **Portuguese (PT)** | 2538 | 846 | 1,246,924 | 58,129 | [24], Sensacionalista, Diário Pernambucano |
| **Spanish (ES)** | 1263 | 421 | 459,406 | 40,891 | [25], FNC |

**Table 3.** Examples of news content for all classes of each language.

| Language | Class | Content |
|---|---|---|
| EN | fake | "Voters on the right have been waiting for this for a long time! Police have finally raided a Democratic strategic headquarters, and the results are devastating! (. . . )" |
| | legitimate | "The search warrant that authorized the FBI to examine a laptop in connection with Hillary Clinton's use of a private email (. . . )" |
| | satirical | "NEW YORK (The Borowitz Report) Speaking to reporters late Friday night, President-elect Donald Trump revealed that he had Googled Obamacare for the first time earlier in the day. (. . . )" |
| ES | fake | "La Universidad de Oxford da más tiempo a las mujeres para hacer los exámenes (. . . )" |
| | legitimate | "El PSOE reactiva el debate sobre la eutanasia El partido del Gobierno llevará su propuesta para regularla al Pleno la próxima semana y cree que saldrá adelante (. . . )" |
| | satirical | "Mucha euforia ha generado el gran lanzamiento del nuevo reality colombiano "Yo me abro" que se estrena hoy y en el que los participantes demostrarán sus habilidades para escapar de la justicia colombiana. (. . . )" |
| PT | fake | "Ministro que pediu demissão do governo Temer explica o motivo: "Não faço maracutaias. Não tenho rabo preso" (. . . )" |
| | legitimate | "Governo federal decide decretar intervenção na segurança pública do RJ. Decreto será publicado nesta sexta-feira (16), segundo o presidente do Senado, Eunício Oliveira. Decisão foi tomada em meio à escalada de violência na capital carioca. (. . . )" |
| | satirical | "A senadora Kátia Abreu é uma mulher que não tira o corpo fora de polêmicas. Ela dá uma tora de árvore para não entrar numa briga mas derruba uma floresta inteira pelo prazer de não sair. (. . . )" |

The complete dataset is made publicly available for other researchers and practitione (http://www.uel.br/grupo-pesquisa/remid/?page_id=145).

*3.3. Classification Algorithms*

In order to explore the best classification model, different ML approaches were evaluated: *k*-Nearest Neighbors (*k*-NN) [45], Support Vector Machine (SVM) [45], Random Forest (RF) [46], and Extreme Gradient Boosting (XGB) [47]. These algorithms were selected due to high predictive performance and broad use in related papers. Linear SVMs were used in Reference [22,24,25], while *k*-NN was used in Reference [48]. Considering ensemble methods, RF and XGB were evaluated in Reference [24,49,50].

Most hyperparameters were kept with default values for the used implementation, which were `scikit-learn` [51] and `xgboost` [47] Python packages. Only the indispensable hyperparameters were chosen, such as *k* for *k*-NN and the kernel of SVM. For *k*-NN, the *k* value was set to 10 after an exhaustive grid search [52], while the distance metric was kept as the default Euclidean distance. On SVM, a linear kernel was used to evaluate the linear separability of the samples, and the other hyperparameters were kept with default values. Moreover, for RF and XGB models, where both algorithms make use of an ensemble approach, the number of subtrees that the model generates was set to 500 to get stable results.

After extracting the feature vectors and removing the outliers, the four classification algorithms were tested through a *k*-fold cross-validation. This process consists in randomly splitting the dataset into *k* groups, holding out the first group for validation, and fitting the model on remaining $k-1$ set [53]. A $k=5$ was used, so for each iteration, 80% of the randomly sampled dataset was used for training and 20% for validation. This process was repeated 20 times, totaling 100 results for each algorithm.

## 4. Analysis and Discussions

The first analysis was conducted to evaluate our hypothesis that language-independent features could be used to identify fake, satirical and legitimate news. With this goal, we applied the Principal Component Analysis (PCA) [54] dimensionality reduction technique for visualization purposes. This way, we can evaluate how well the proposed features model each class behavior. Figure 2 illustrates news articles distribution over a two-dimensional feature space, projected by components 1 and 2 from PCA. The projection exposes a particular behavior onto all languages, where Principal Component 1 (more than 20.00% variance explained) reflected a tendency of separation among the classes, as expected. The clearest separation between classes is observed in PT news articles, as seen in Figure 2b. On the other hand, EN news distribution has more overlaps (Figure 2a), followed by Spanish news (Figure 2c), i.e., the separation between classes is less straight forward. For all three languages, the legitimate class is positioned on the positive side of the X-axis, indicating that there is similarity in classes distribution over different languages. Fake and satirical classes were positioned on the left of the chart in all plots, with satirical samples being less scattered than fake ones. Although fake and satirical classes appear to have a more difficult distinction, it is remarkable that their position on the feature space is similar across sets. This analysis confirms the hypothesis that our language-independent features can model news behavior from different languages. PCA results corroborated with our conjecture towards the use of ML models to detect news intent between fake, satirical and legitimate classes.

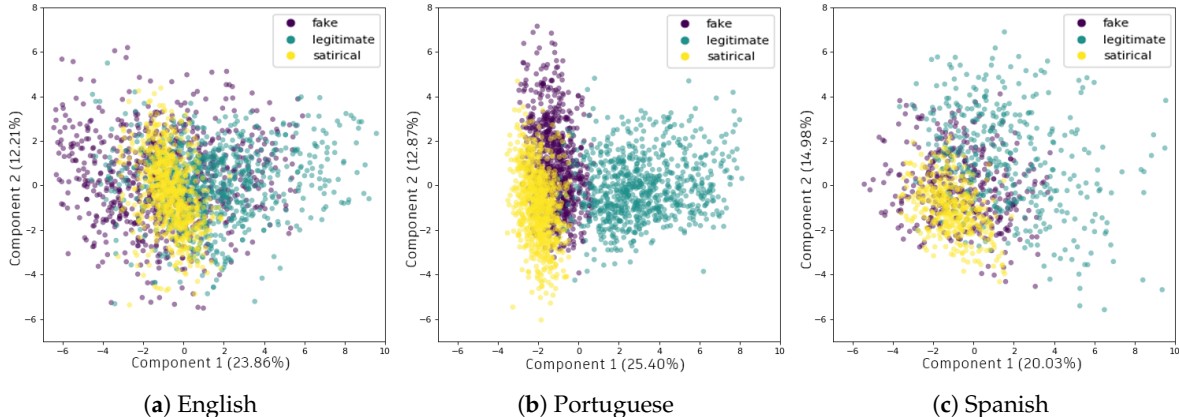

**Figure 2.** Two-dimensional plot of Principal Component Analysis (PCA) of three datasets showing the distribution of classes on a lower dimensional vector space.

The models' accuracies are plotted through boxplots on Figure 3, showing the distribution of results with the central tendency being the median. The results show that *k*-NN was the worst algorithm regarding predictive power, with a mean accuracy of 75% (EN), 89% (PT), and 75% (ES). The ensemble algorithms, RF and XGB, achieved similar and stable results, with RF reaching 79.9% (EN), 93.9% (PT), and 82% (ES); and XGB achieving 80.3% (EN), 94.7% (PT), and 82% (ES). SVM had the second worst predictive performance on EN (79%), the best performance on PT (95%) while tying with ensemble algorithms on ES (82%). Results indicate that PT and ES collections have more linear separability comparing to EN, with a slightly more linear behavior on PT.

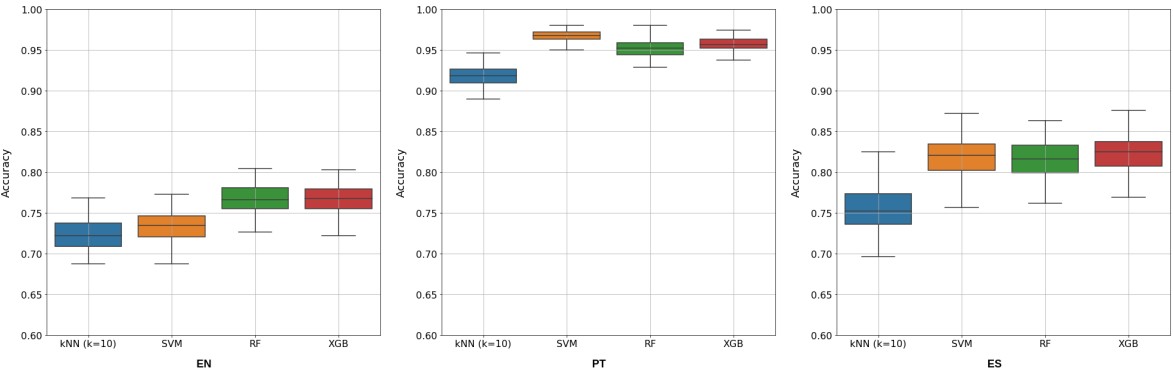

**Figure 3.** Results of cross-validation for each algorithm (*k*-Nearest Neighbors (*k*-NN), Support Vector Machine (SVM), Random Forest (RF), and Extreme Gradient Boosting (XGB)) showing accuracy distribution for all three datasets (English, Portuguese, and Spanish).

The good predictive performance achieved by all algorithms demonstrates that the language-independent features and pipeline proposed in this study are efficient to identify fake, legitimate, and satirical news. Since the three classes are balanced on all datasets, the baseline classification accuracy is 33%. Therefore, when using the set of features proposed in this paper, the average accuracy of models reaches 84%, 2.5 times better than the baseline.

To compare the models induced by the algorithms and evaluate which algorithm generated the best models, we employed the Friedman Statistical Test (FST). FST is a statistical test that ranks multiple methods over several datasets, as described in Demšar [55]. In FST, the null hypothesis to be tested is that all algorithms performed equally well, i.e., whether there is a significant difference among the results. If the null-hypothesis is rejected, the Nemenyi post-hoc test is used to compare the classifiers, and their performances are considered significantly different if the corresponding average ranks differ by a Critical Difference (CD) metric. For this study we use a confidence level of $\alpha = 0.05$.

Figure 4 presents the Nemenyi post-hoc test, showing that RF was the best performing algorithm, followed by XGB and SVM. However, the distances within these algorithms are less than the CD. Thus, statistically speaking, the algorithms have similar performances, which is on pair with the results shown in Figure 3. Differently, the classifiers generated by *k*-NN presented a distance higher than the CD when compared to both RF and XGB, meaning that it was the worst performing algorithm from the set. However, note that *k*-NN performance does not differ statistically from the SVM. The random baseline (RND) was also included in the test, which shows that all classifiers being tested had a statistically better performance comparing to the baseline. This analysis also confirms the power of language-independent features as news descriptors, as the high predictive performances do not rely on classifiers capabilities but on the features quality.

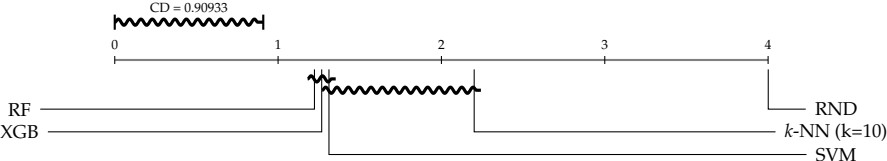

**Figure 4.** Comparison of all classifiers against each other with the Nemenyi test considering its accuracy results. The connected group of classifiers are not significantly different (at $\alpha = 0.05$).

Regarding performance by class, Figure 5 presents the normalized confusion matrix summing results from the cross-validation process. The matrices show the percentage of a predicted label versus the true label of samples, indicating the behavior of classifiers for each class. Thus, it is possible to identify the most challenging type of news to be detected.

From results, it is clear that fake documents are the most difficult to identify. This is reasonable since the fake class is overlaid by other classes and is very scattered, especially on EN and ES dataset, as seen on PCA feature space in Figure 2. This may be explained by the deceptive nature of fake news, where the intention is to make the content look like a legitimate article. In this context, for both EN and ES datasets, fake was mostly misclassified as legitimate, highlighting the deceiving characteristic of fake news. Contrarily, in the PT dataset, fake was mainly misclassified as satirical.

The legitimate class was the second most accurate on all corpora, being mostly mistaken as a fake in all three languages. This can be explained by the similarity between both classes. Moreover, a fake document main goal is to simulate a legitimate behavior, which deceives models in this direction. When looking at the PCA feature space, both classes share a similar scattered space, making the models misclassify a sample as belonging to another class.

Figure 5 also shows that the satirical class was the most easily separable from the others. This result is on pair with PCA in Figure 2, where the samples from this class are on a denser area, i.e., they are closely grouped in the feature space. Satirical news was misclassified almost equally as fake and legitimate by the classifiers on EN and PT, with a tendency to be predicted as fake on ES corpus.

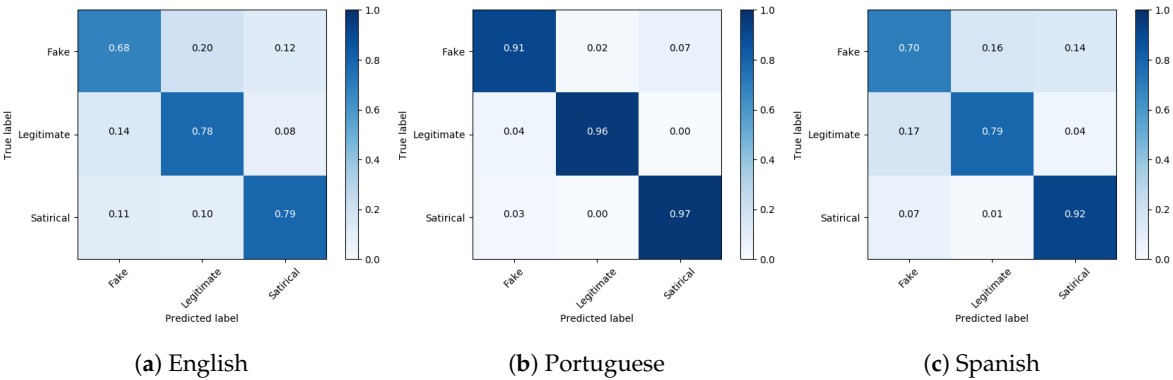

**(a)** English        **(b)** Portuguese        **(c)** Spanish

**Figure 5.** Normalized confusion matrices comparing the true label of samples with the predicted label along the cross-validation process. The matrices indicates the class that was most wrongly predicted by models and the class which models was mistakenly towards to predict.

When comparing with the other studies, the present work achieved either the same or higher results regarding performance on detecting fake, legitimate, and satirical news. In Horne and Adali [22], the results are evaluated with all combinations of classes (fake vs. real, satire vs. real and satire vs. fake), reaching between 71% and 91% accuracy scores, depending on the combination, which is relatable with the 80% we achieved on EN dataset. However, the authors do not consider a multi-language scenario. The works of Monteiro et al. [24] and Posadas-Durán et al. [25] achieve a maximum accuracy of 89% and 77%, respectively, on binary classifications (fake and legitimate). Our method improves previous studies by achieving 95% and 82% accuracies with three classes of news (fake, legitimate, and satirical).

It is important to highlight that Monteiro et al. [24] tested both content specific features, like Bag-of-Words and stylometric features. Posadas-Durán et al. [25] used only Bag-of-Words and POS-tag *n*-grams, which differs from the approach of language-independent features we proposed. However, our results using stylometric features achieved better performance in predicting news classes in comparison to both works, indicating that the language-independent approach we presented may lead to better modelling of the problem of fake news detection.

To understand the importance of language-independent features used, FST was conducted to compare the feature sets for all languages: *Complexity*, *Stylometric*, and *Psychological*. Each combination within the three feature sets was analyzed, totaling 6 possible combinations for each of the 3 languages. The best performing algorithm from the former analysis (RF) was used to induct the models, ranking the performance by class using the *F*1-score metric.

The results presented in Figure 6 shows that the combination *Complexity + Stylometric* features had the best performance, followed closely by the combination of all features. Next, without statistical difference from the first two, is the set of *Stylometric + Psychological* features, which had no statistical difference from using only *Stylometric* features. The *Complexity* set and the *Complexity + Psychological* combination come next, with a statistically significant difference from the others. The *Psychological* feature set was the least performing one with statistical difference from all other sets.

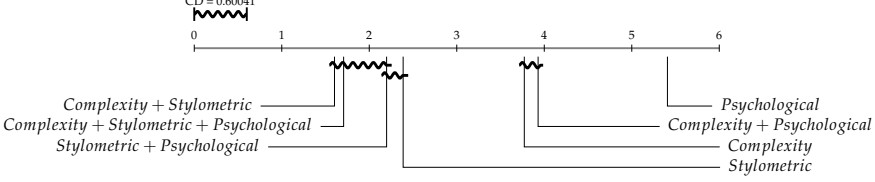

**Figure 6.** Nemenyi post hoc test for all feature category combinations.

While the *Complexity + Stylometric* and *Complexity + Stylometric + Psychological* are tied in the first place, with no significant difference because they are within the Critical Difference, it is possible to discard the *Psychological* set in this situation, considering that removing features makes the problem modelling less complex. The *Complexity* set demonstrated a good predictive power when combined with *Stylometric*, but using the set alone or with *Psychological* does not improve the performance of the model significantly.

Going even further, considering just the *Stylometric* set of features is not enough to solve the problem, using only 16 of the 21 features the model can still perform the same as *Stylometric + Psychological*, which do not differ significantly from *Complexity + Stylometric* and *Complexity + Stylometric + Psychological*. The *Psychological* feature demonstrated a low predictive contribution, probably because we used only one feature of this category in this work. Different than Horne and Adali [22], which used dictionary features in this category (e.g., the number of analytic words) we focused on features that could be assessed by all languages evaluated, thus, considering that some languages have fewer resources than others [56], we used the polarity feature which is widely available [40].

Table 4 exposes the average numerical value of each feature for each class grouped by corpus. Therefore, by analyzing the central tendency for each class through all corpora, we can discuss patterns that are present across languages, e.g., if some feature happens to have a higher value for legitimate, followed by fake and satirical for all corpora. A similar discussion is also made in Horne and Adali [22] and Gruppi et al. [29].

**Table 4.** Average values of all features grouped by class and by language corpus, in alphabetical order.

|  | EN | | | PT | | | ES | | |
|---|---|---|---|---|---|---|---|---|---|
|  | Fake | Legitimate | Satire | Fake | Legitimate | Satire | Fake | Legitimate | Satire |
| avg_word_size | 4.40 | 4.30 | 4.18 | 4.14 | 4.28 | 4.38 | 4.30 | 4.35 | 4.40 |
| entities_ratio | 0.007 | 0.008 | 0.009 | 0.013 | 0.011 | 0.009 | 0.008 | 0.009 | 0.005 |
| oov_ratio | 0.100 | 0.094 | 0.009 | 0.062 | 0.050 | 0.079 | 0.023 | 0.019 | 0.036 |
| polarity | 0.011 | −0.075 | 0.010 | −0.262 | −0.303 | −0.245 | −0.411 | −0.388 | −0.299 |
| pos_diversity_ratio | 0.09 | 0.05 | 0.07 | 0.44 | 0.22 | 0.50 | 0.21 | 0.15 | 0.20 |
| quotes_count | 8.14 | 23.43 | 7.26 | 5.00 | 6.80 | 6.06 | 5.29 | 8.90 | 0.01 |
| quotes_ratio | 0.002 | 0.004 | 0.003 | 0.003 | 0.002 | 0.003 | 0.005 | 0.001 | 0.000 |
| ratio_ADJ | 0.078 | 0.082 | 0.081 | 0.055 | 0.057 | 0.065 | 0.041 | 0.044 | 0.045 |
| ratio_ADP | 0.101 | 0.112 | 0.110 | 0.121 | 0.140 | 0.128 | 0.144 | 0.155 | 0.153 |
| ratio_ADV | 0.042 | 0.045 | 0.049 | 0.039 | 0.028 | 0.036 | 0.038 | 0.037 | 0.051 |
| ratio_DET | 0.078 | 0.082 | 0.089 | 0.101 | 0.105 | 0.111 | 0.123 | 0.125 | 0.130 |
| ratio_NOUN | 0.185 | 0.178 | 0.175 | 0.168 | 0.174 | 0.193 | 0.164 | 0.180 | 0.173 |
| ratio_PRON | 0.028 | 0.029 | 0.031 | 0.053 | 0.039 | 0.053 | 0.034 | 0.034 | 0.041 |
| ratio_PROPN | 0.11 | 0.10 | 0.10 | 0.11 | 0.10 | 0.09 | 0.08 | 0.11 | 0.06 |
| ratio_PUNCT | 0.124 | 0.128 | 0.109 | 0.102 | 0.101 | 0.103 | 0.149 | 0.137 | 0.132 |
| ratio_SYM | 0.003 | 0.001 | 0.004 | 0.011 | 0.012 | 0.011 | 0.012 | 0.026 | 0.001 |
| ratio_VERB | 0.149 | 0.152 | 0.160 | 0.095 | 0.078 | 0.088 | 0.126 | 0.117 | 0.139 |
| sentences | 34.0 | 49.9 | 25.1 | 10.3 | 17.2 | 12.0 | 13.2 | 67.3 | 8.1 |
| ttr | 0.522 | 0.439 | 0.491 | 0.596 | 0.412 | 0.669 | 0.532 | 0.457 | 0.562 |
| upper_case | 132 | 199 | 90 | 66 | 126 | 31 | 41 | 212 | 28 |
| words_per_sents | 19.7 | 25.1 | 22.7 | 17.9 | 20.6 | 27.5 | 37.9 | 36.6 | 31.0 |

The following features present the same pattern on all three corpus: `ratio_ADP`, `ratio_DET` and `upper_case`. These features have the same order of average value by class regardless of the language being evaluated. That is, the class with the highest value for a feature, the second and the last one are the same, maintaining the order, on all corpus in this work. For example, for the `upper_case` feature, the order of *legitimate > fake > satire* is maintained on all three corpora being analyzed.

The Type-Token Ratio (TTR) values on all three corpora are lower for legitimate news, followed by fake or satirical. Since TTR measures the lexical diversity of text vocabulary, this may lead to the belief that legitimate news has a poorer vocabulary. However, we must consider that morphological complexity may change depending on the language, where languages with greater complexity may affect the result obtained in TTR. Nonetheless, in this case, this is probably due to the size of the text, because fake and satirical news tends to be smaller on all corpora. For all three corpora, the same behavior was observed in all types of news, meaning that language complexity did not affect analysis. An analogous phenomenon was observed in both Reference [22,24].

Some other features are noteworthy due to the pattern found in PT and ES corpora, which may be explained by similarities between these languages [57]. The average word size (`avg_word_size`) is an example of this pattern. On EN the highest values are from the fake class, followed by legitimate and satirical classes, but the behavior is reversed on PT and ES. The ratio of OOV words (`oov_ratio`) follows the order *fake > legitimate > satire* on EN, but *satire > fake > legitimate* on PT and ES. Similar patterns are also found on `ratio_ADJ` and `ratio_SYM` features.

Finally, regarding the proposed features, i.e., `pos_diversity_ratio`, `quotes_ratio`, `oov_ratio`, and `entities_ratio`, it is important to mention their importance ranking as 1st, 6th, 9th, and 11th positions, respectively, as Figure 7 shows. The high placement in this rank means that they actively affect the predictive performance, showing that our proposed features are decisive in a multi-language scenario.

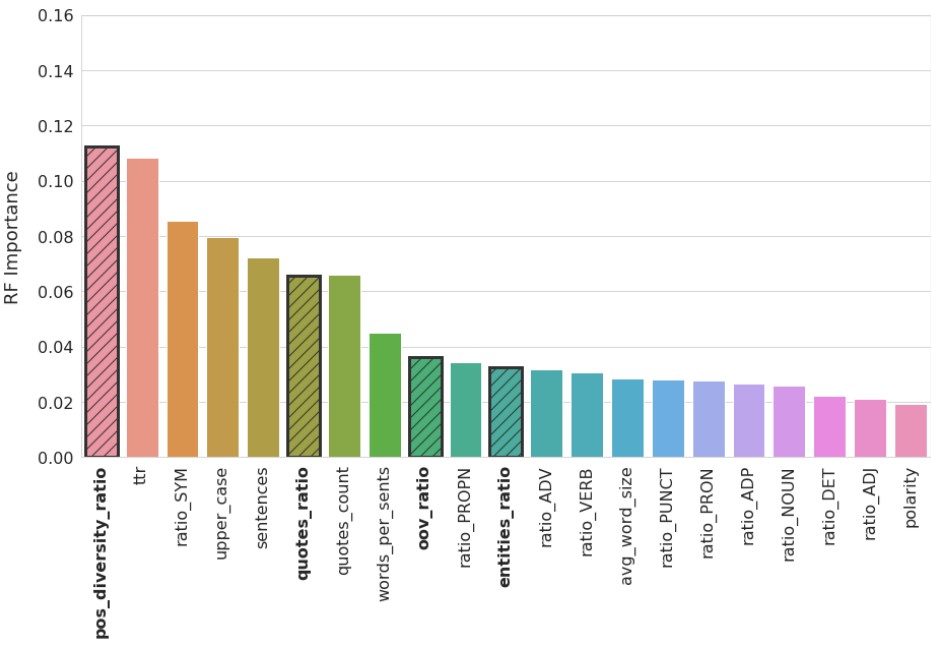

**Figure 7.** Features sorted by their normalized importance computed using RF importance. Proposed features are highlighted in bold.

## 5. Conclusions

The identification of deceptive news is required due to the increase of news consumption through OSM, which provides unruled content broadcasting. Moreover, the spreading speed in OSM requires an automated method to accomplish the detection task. In this paper, we presented a comparison of language-independent features of stylometric, complexity, and psychological types in a multi-language scenario. Furthermore, we proposed stylometric features to improve the identification of satirical, legitimate and fake news articles over three different languages: American English, Brazilian Portuguese, and Spanish. The first contribution of this paper is the creation of a curated

multi-language corpora composed of news articles from three different classes. Moreover, we made this corpus available for other researchers and practitioners.

To detect news intent, we presented a text processing pipeline under content-based premises. The preprocessing stage is composed of filtering, cleaning and noise removal phases. Then, *Complexity*, *Stylometric*, and *Psychological* features are extracted from textual data. Moreover, feature importance was explored toward supporting a suitable predictive ML model using 9930 news articles. Support Vector Machine (SVM), *k*-Nearest Neighbors (*k*-NN), Random Forest (RF), and Extreme Gradient Boosting (XGB) were compared to recommend one that best fit the detection model.

The RF algorithm reached the highest performance, achieving an average accuracy of 85.3%, followed closely by XGB and SVM, with no statistical difference. The overall performance of ML models indicates that purely stylometric features, including the proposed POS-tag diversity, the ratio of named entities to text size, the ratio of quotation marks to text size, and the OOV words frequency, were capable of enhancing the predictive outcomes, being statistically superior to the other compositions with RF, XGB, and SVM as the most predictive algorithms.

Besides that, the shared pattern between the studied languages suggests there is an underlying behavior among different languages, which can support fake news detection over several idioms beyond those explored in this work.

For future works, we will evaluate transfer learning strategies, using pre-trained models to extract more abstract features, such as semantic level characteristics. Either pre-trained word embeddings can be assessed using the multi-language corpora assembled in this paper.

**Author Contributions:** All the authors contributed equally to this work. All authors have read and agreed to the published version of the manuscript.

**Funding:** This study was financed in part by the Coordenação de Aperfeiçoamento de Pessoal de Nível Superior - Brasil (CAPES) - Finance Code 001, Coordination for the National Council for Scientific and Technological Development (CNPq) of Brazil - Grant of Project 420562/2018-4 and Fundação Araucária (Paraná, Brazil).

**Conflicts of Interest:** The authors declare no conflict of interest.

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
