# Peer review of "Language-Independent Fake News Detection: English, Portuguese, and Spanish Mutual Features"

_futureinternet, doi:10.3390/fi12050087_

Round 1
Reviewer 1 Report
This paper proposes a language-independent fake news detections mechanism for English, Portuguese and Spanish languages. It compares language-independent features of these languages to detect news considering fake, satirical ad legitimate classes.
This is a good piece of scholarship. I could not identify any flaws that need to be corrected.
Author Response
Dear Reviewer, you can find the response letter attached.

Reviewer 2 Report
The paper aims at identifying and analyzing the contribution of language-independent features in detecting fake, legitimate or satyrical news. More in particular, it analyses a corpus of news written in English, Spanish and Brazilian Portuguese.
The paper is well written and structured, and the content is sound.
The introduction section successes in providing the reader with the necessary context.
The second section provides a reach enough review of related works, highlighting how the paper differs from the others.
The third section presents the text processing pipeline exploited by authors for analyzing the language-independent features in news detection. Then the dataset analyzed is described, which is also made available to the research community by the authors. Finally the set of (classic) classification algirthm exploited is presented.
The results provided and their analysis seem to be rich and generally sound.
Some minor fixes follow;
row 74: purely NLP classifier -> pure NLP classifier
row 79: Spanish speaking -> Spanish-speaking
rows 100 and 106: f alse -> false
row 232: disconsider -> disregard
row 243: a corpora -> a corpus [or simply corpora] (corpora is a plural noun)
row 284: choosen -> chosen
row 349: overlayed -> overlaid
row 443: this corpora -> this corpus (corpora is a plural noun)
Author Response

(The authors gave the same response as above.)

Reviewer 3 Report
The authors present a text processing pipeline for classifying articles, over three languages (English, Portuguese, and Spanish), into three classes: satirical, legitimate, and fake-news. To this end, they suggest language-independent features and evaluate their predictive power using PCA, a well-known machine learning classifiers, and post-hoc Nemenyi test.
This paper addresses the problem of fake news detection, which has become an emerging topic of great importance in recent social media research. The paper is well written and technically focused: it will be of interest to the readers without any doubt.
This reviewer recommends the paper for publication with the following minor changes:
1) In section 2 (Definitions and Related Work), the authors should examine also the recent work by Guibon et al. (Guibon, G., Ermakova, L., Seffih, H., Firsov, A., Le Noé-Bienvenu, G. Multilingual Fake News Detection with Satire. 2019, April.);
2) Do the three datasets EN, ES, PT, include the same news, translated into respective languages? To what extent does this influence the poor class separation shown in Figure 2? This point should be further clarified.
3) It is not clear to what language dataset (EN, PT, EN) the Nemenyi test reported in Figure 6 is referred to. The same test should be applied to each language dataset and related results compared.
4) P9, L305: "EN news distribution has more overlaps (Figure 2c)" should read "ES news…"?
Author Response

(The authors gave the same response as above.)

Reviewer 4 Report
This paper presents a new approach to classify fake news. The proposed method is based on textual features and can be implemented independently of the language.
This paper is very interesting. The proposed methodology is clear and described in adequate detail. The figures are adequate, and the experimental results show that the proposed method has good precision in detecting false and satirical news in several benchmark data sets. In general, the scientific quality of the paper and its relevance in the field of computing is good.
However, from my point of view, the feature extraction phase used in its methodology is underestimated. Authors should improve the description of the extracted features. This is the most important part of this research work and would help to better understand the behavior of classifiers based on these characteristics.
Author Response

(The authors gave the same response as above.)
